# The Role of Autophagy in the Mineralization Process of Bone and Dentin

**DOI:** 10.3390/ijms26136278

**Published:** 2025-06-29

**Authors:** Ian Moran, Cassandra Villani, Anne George

**Affiliations:** Brodie Tooth Development Genetics & Regenerative Medicine Research Laboratory, Department of Oral Biology, University of Illinois at Chicago, Chicago, IL 60612, USA; itrooper.m@gmail.com (I.M.); cvilla61@uic.edu (C.V.)

**Keywords:** autophagy, mineralized tissue, bone, dentin, mineralization, extracellular matrix

## Abstract

Autophagy is a cellular process that recycles intracellular macromolecules and degrades toxic cytoplasmic material to provide the cell with nutrients and facilitate survival. Although autophagy and its role in the differentiation of osteoblasts, osteoclasts, and odontoblasts has been described, the importance of autophagy during matrix mineralization remains unaddressed. This review aims to characterize the autophagy/matrix mineralization relationship and elucidate the significance of autophagy during matrix mineralization. During the mineralization process, autophagy is important for cell survival and promotes the differentiation of osteoblasts and odontoblasts, the key cells that facilitate bone and dentin formation. Differentiation of these cells results in the synthesis of an organic proteinaceous matrix which subsequently forms the template for the deposition of calcium and phosphate to ultimately form crystalline hydroxyapatite. In bone, autophagy influences osteoblastic/osteoclastic activity and bone remodeling. In dentin, autophagy participates in odontogenic differentiation and facilitates odontoblastic secretion of dentin matrix proteins. This review aims to show that autophagy is critical for bone mineralization and tooth formation by supporting intracellular signaling pathways required for cell differentiation and subsequent matrix mineralization.

## 1. Introduction

Autophagy is a cellular process that recycles intracellular macromolecules and degrades toxic cytoplasmic material to provide the cell with nutrients and facilitate cell survival [1,2,3]. This process is induced by stressful cellular conditions including nutrient deprivation and exposure to drugs, such as rapamycin, and reactive oxygen species [2,4]. In response to stress, the unc-51 like autophagy activating kinase 1 (ULK1) complex is activated leading to the activation of autophagy. This can be negatively regulated by direct phosphorylation of ULK1 via mTORC1 [5]. In pre-osteoblasts and pre-odontoblasts, stressful conditions elicit an increase in cytosolic calcium [6]. The cellular processes required for the differentiation of these cells are dependent on this intracellular calcium increase [6]. Once autophagy induction occurs, an autophagosome (a double membrane vesicle) is produced to transport intracellular material to the lysosome for degradation [2,7]. At the lysosome, the autophagosome fuses with the lysosome, forming an autolysosome where lysosomal hydrolases degrade the intracellular material [8]. The autolysosome then remodels itself into a lysosome which can participate in autophagy again (Figure 1A) [2,7,8].

There are three main types of autophagy: macroautophagy, chaperone-mediated autophagy, and microautophagy [7]. Macroautophagy (the focus of this review; hereinafter called “autophagy”) occurs via lysosome–autophagosome fusion and includes five steps: initiation, phagophore nucleation, phagophore elongation, fusion, and degeneration (Figure 1B) [7]. Chaperone-mediated autophagy occurs via lysosomal degradation and selectively targets KFERQ-like motif-bearing proteins (Figure 1C) [9]. In microautophagy, cytoplasmic materials or KFERQ-like motif-bearing proteins are degraded directly via lysosomal or endosomal invagination (Figure 1D) [7,9]. Outside of the three main autophagy types, mitophagy is the autophagy of the mitochondrion and is used to aid mitochondrial metabolism and to eliminate damaged mitochondria [10]. Mitophagy is important in osteoclastic differentiation and maturation since osteoclasts contain many mitochondria to meet their high-ATP demand [10].

Although autophagy is predominantly known as a degradative process, studies have shown secretory autophagy to be a method of unconventional protein secretion (UPS) [5]. Conventional protein secretion requires the endoplasmic reticulum (ER) and Golgi to secrete cargo with an N-terminal leader peptide in COPII+ vesicles via exocytosis while UPS includes alternate pathways for leaderless proteins [5,11]. There are three stages to secretory autophagy: (1) recruitment of cargo to the forming autophagosome; (2) transport of the cargo–receptor complex to the membrane; (3) internalization of the cargo to the autophagosome and fusion to the plasma membrane [12]. The fusion of the autophagosome to the plasma membrane and cargo selection require specialized mechanisms compared to macroautophagy, however, the cellular machinery for this process overlaps with that for conventional protein secretion and exosome biogenesis [5].

Autophagy plays a role in multiple biological processes such as metabolism, embryogenesis, cellular differentiation, organ development, mineralization, aging, and immunity [1,7,13,14]. Due to the key role of autophagy in different cell types, autophagic malfunctions can affect various organs and tissues. These malfunctions have been associated with diseases including cancer and heart disease [7]. Under homeostatic, non-disease conditions, autophagy is known to participate in bone mineralization through the presence of intracellular mineral structures within autophagic-like vesicles [15]. However, since autophagy can be modified by hormonal and soluble signals that influence bone mineralization, conditions that disrupt the intersection of autophagy and bone mineralization signaling process can give rise to bone disorders like osteopenia, osteopetrosis, or osteoporosis [9].

Outside of mineralization, autophagy is a process critical to maintaining cellular function. In bone, autophagy serves to regulate bone mineralization and support osteoblast and osteoclast differentiation [16,17]. In dentin, autophagy stimulates dental pulp stem cell survival and promotes odontoblast differentiation in cells exposed to inflammation [18,19]. In this review, we aim to demonstrate the importance of autophagy during matrix mineralization.

## 2. Autophagy and Matrix Mineralization

Mineralization is the cellular process of calcium phosphate crystal deposition in the extracellular matrix (ECM) that forms hard tissues such as bone and teeth [20]. Previous studies, such as the one conducted by Su et al., demonstrated that autophagy facilitates osteoblast mineralization and controls bone homeostasis [21]. Their study, using primary murine cells, found that osteoblasts release matrix vesicles (MVs) during mineralization and that AnxA5, a phospholipid binding protein within the annexin A protein family, is involved with MV-mediated mineralization of bone tissue and osteogenic differentiation (Figure 2A,B) [21]. Adherent MVs attach to the ECM containing type I collagen, via the AnxA5 protein, allowing these vesicles to stimulate osteogenic progenitor cells and support mineralization [21]. Additionally, MVs, via AnxA5, were shown to facilitate autophagy in bone marrow mesenchymal cells [21]. However, when autophagy decreased, osteoporosis occurred and AnxA5 expression decreased [21]. In a similar study, Wang et al. found that decreased autophagic activity hinders osteoblast mineralization [22]. This effect was shown to be a result of inhibited osteoblast differentiation and apoptosis occurs due to increased oxidative stress [22].

In teeth, MV-mediated mineralization occurs in dentin and cementum [23]. During MV-mediated mineralization, calcium phosphate ions form inside MVs and precipitate into hydroxyapatite crystals. Once the crystals are exported from the MVs into the ECM they are positioned around the collagen fibers produced by odontoblasts (Figure 2C) [23]. Li et al. demonstrated that autophagy is critical for mineralization, and Couve et al. suggested that autophagy plays a role in the differentiation of immature odontoblasts due to the detection of microtubule-associated protein light chain 3 (LC3), an autophagosome marker [19,24,25]. Additionally, when transitioning from primary dentinogenesis to secondary dentinogenesis, odontoblasts, via autophagy, decrease the amount of secretory machinery and begin to solely secrete secondary dentin [25]. While the formation of mineral is known to take place within MVs, it is still unclear which cellular processes contribute the formation of these vesicles [26,27]. However, autophagy has still been shown to be important for appropriate odontoblast function and the mineralization process due to its role in various mechanisms of vesicle formation.

**Figure 2 ijms-26-06278-f002:**
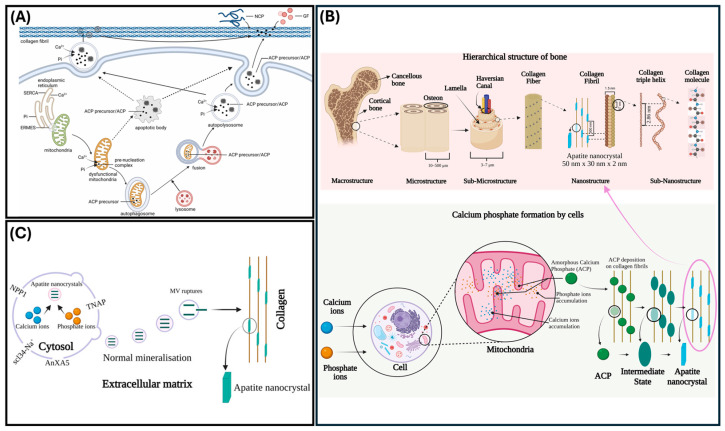
Mechanisms of matrix mineralization. (**A**) Biomineralization model demonstrating the various mechanisms through which amorphous calcium phosphate is formed (reprinted with permissions from Yan et al., 2023 [28]). (**B**) Bone anatomy hierarchy and ACP production and secretion to ECM. (**C**) Schematic of cell-dependent biomineralization (reprinted with permissions from Indurkar et al., 2023 [29]).

### Ions Associated with Autophagy

Though magnesium is known to be critical for proper bone health, magnesium acts as an antagonist of calcium in calcium-mediated ECM mineralization [24]. In doing so, magnesium inhibits mineralization by decreasing the calcium concentration in mitochondria and by decreasing intracellular calcium [24]. Due to this decrease in mitochondrial calcium, MVs cannot be formed or released [24]. For MVs that are formed, increases in Mg^2+^ can also decrease mineral nucleation within the vesicles and alter the resulting mineral structure [24]. Aside from its effect on MVs, increased levels of Mg^2+^ also decreases the production of collagen I at both the mRNA and protein levels [24]. While increased Mg^2+^ can affect bone quality, low Mg^2+^ levels can also result in increased bone resorption potentially through increased inflammatory responses [30]. This bidirectional effect highlights not only the importance of magnesium in bone homeostasis but indicates that further study on its effect on autophagic processes is important for understanding the intersection of its physiological and pathological functions.

A subsequent effect of autophagy inhibition within osteocytes is low bone mass [24]. Thus, Li et al. posited that autophagy is a critical process for mineralization [24]. Another study by Zhu et al. investigated the relationship between Mg^2+^ stimulated macrophages, exosomal cargo changes, and the osteogenic differentiation of bone marrow stromal stem cells (BMSCs). The group found that Mg^2+^ can induce autophagy in macrophages to influence their polarization and subsequently alter the miRNA cargo of secreted exosomes. The Mg^2+^ -mediated exosomes were then shown to increase the osteogenic differentiation of BMSCs by promoting autophagy [31]. This was shown to be an effect of decreased exosomal expression of miR-381, a miRNA previously demonstrated to inhibit BMSC osteogenic differentiation levels [31].

Other ions associated with autophagy are calcium and iron. Calcium has pro- and anti-autophagic effects [32]. Calcium can stimulate autophagy by engaging the mTOR and AMPK pathways and inhibit AMPK-dependent autophagy by being sequestered in mitochondria during physiological signaling [32]. Calcium is also an important factor in aberrant mineralization as occurs with vascular calcification associated with kidney and cardiac diseases [33,34]. A high concentration of iron, transported via the divalent metal transporter 1 (DMT1) protein, facilitates osteoblast autophagy [32]. Increased DMT1 expression increased the amount of iron in osteoblasts [32]. These studies summarize the multifactorial role ions play in homeostatic mineralization and changes in their concentration can negatively affect health through altered autophagic signaling.

## 3. Autophagy in Bone

Osteoblasts secrete the components required for bone formation and with time can be encapsuled within the bone becoming osteocytes, which play an integral role in bone maintenance. Osteoclasts oppose osteoblast function and differentiate from hematopoietic cells to resorb bone [15,32,35]. According to Shapiro et al., the role of autophagy in osteoblasts was unknown [35]. Nollet et al. demonstrated that autophagy participates in osteoblastic mineralization as shown in Figure 3A [15]. Nollet’s group found that *Atg7* repression resulted in a decrease in mineralizing nodules within osteoblasts and *Becn1* repression decreased osteoblastic mineralization capacity [15]. Nollet et al. concluded that a decrease in autophagic activity would limit osteoblastic mineralization capacity [15]. A study on the osteogenic differentiation of MC3T3-E1 cells recently demonstrated that CYB5A, a cytochrome important for various redox reactions, stimulated autophagy through the AKT/mTOR/ULK1 pathway and promoted differentiation [36]. Continued exploration of this protein in other cell lines will be valuable for confirming the intersection of osteogenesis and autophagy.

Another important gene for autophagy in murine osteoblasts is *Rubcn*, which encodes for the protein Rubicon and is responsible for creating autophagosomes and degrading autophagic cargo via lysosomes [37]. Yoshida et al. found that *Rubcn* knock-out in osteoblasts stimulates osteoblast function and increases autophagy [37]. Yoshida et al. explained that the inactivation of Rubicon increases mineralization through increased osteoblast differentiation and autophagy activity [37]. *Rubcn* knock-out in osteoblasts may also increase osteocyte differentiation since there was an elevated level of the osteocyte marker, PDPN/E11, in primary osteoblasts [37].

Due to their energy-intensive activity, osteoclasts contain many mitochondria so that ATP molecules can be utilized for bone resorption [10]. These mitochondria facilitate osteoclastic activity, differentiation, and maturation [10]. During differentiation and maturation, osteoclasts demonstrate an increase in autophagy (Figure 3B) [10]. Aoki et al. demonstrated that autophagy directs the maintenance of mitochondrial function in osteoclasts [10].

The ratio of osteoblasts and osteoclasts, and, by extension, bone remodeling, is dependent on the regulation of bone cell survival/apoptosis [15]. Since autophagy is involved in cell survival, autophagy may affect the formation and resorption of bone (Figure 3C) [15]. When Onal et al. deleted the *Atg7* gene within murine osteocytes, another important cell type in bone, autophagy was suppressed [38]. At six weeks, male knock-out mice demonstrated a decrease in bone mineral density, whereas female knock-out mice did not exhibit changes in bone mineral density anywhere on their skeleton [38]. At six months, both male and female knock-out mice demonstrated low bone mass at all skeletal sample locations [38]. Once the murine skeletons were analyzed, Onal et al. discovered a decrease in spongey bone volume at the spine and femur, a decrease in cortical bone thickness, and an increase in cortical bone porosity [38]. These results led Onal et al. to conclude that autophagy suppression in the osteocytes of young mice resulted in skeletal changes characteristic of older mice [38]. Despite the potential autophagic effect on bone remodeling via cell survival/apoptosis, Onal et al. did not find changes in osteocyte apoptosis when *Atg7* was deleted [38]. Osteocyte apoptosis is correlated with osteoclast formation; an increase in osteocyte apoptosis would not explain Onal et al.’s discovery of decreased osteoclast formation in *Atg7* knock-out mice [38]. The authors also noted that estrogen deficiency has been associated with an increase in oxidative stress and bone loss [38]. Previous studies have shown age-related decreases in bone mineral density and rate of bone formation/remodeling that are associated with increased reactive oxygen species (ROS) levels [39]. The study by Almeida et al. indicated that the effects on bone caused by increased oxidative stress can be accelerated by sex hormone deficiencies by limiting cellular defenses against ROS [39]. A more recent study using human mesenchymal stem cells (MSCs) was able to demonstrate that during osteoblast differentiation estrogen upregulates RAB3GAP1, part of a complex that plays a role in autophagosome biogenesis [40]. Together, these studies suggest a complex role for estrogen and other sex hormones in the regulation of bone homeostasis through osteoblast cell survival by modulating ROS and autophagy [38,39,40].

**Figure 3 ijms-26-06278-f003:**
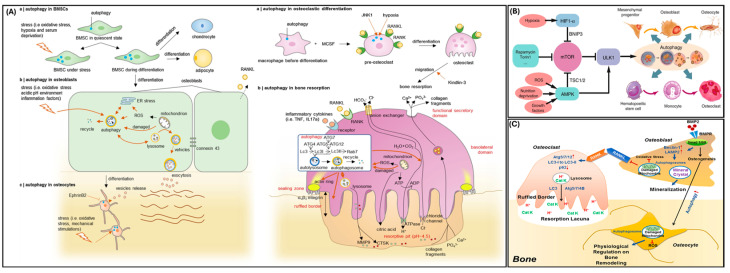
Autophagy in bone. (**A**) Autophagy cellular mechanism showing the role of autophagy in osteoblasts and osteoclasts (reprinted with permissions from Li et al., 2020 [41]). (**B**) Autophagy signaling pathways that regulate autophagy in bone (reprinted with permissions from Yin et al., 2019 [42]). (**C**) Role of autophagy in differentiation of osteoblasts, osteoclasts, and osteocytes and its regulatory its role in bone remodeling (reprinted with permissions from Xiao et al., 2019 [43]).

## 4. Autophagy in Odontoblasts

Odontoblasts are dentin-secreting cells located at the dentin–pulp complex in mammals [25]. These cells are long-lived and are not usually replaced during an organism’s life [25]. While odontoblast cell bodies reside in the deepest aspect of dentin, an odontoblastic process projects from each cell body into dentin via a dentinal tubule [44]. In conjunction with the maxillary and mandibular branches of cranial nerve V, odontoblasts can recognize changes in temperature, mechanical force, and pain [25].

Dental pulp cells (DPCs) are located in the dentin–pulp complex and differentiate into odontoblast-like cells that secrete dentin [45]. Within DPCs, autophagy participates in odontoblastic differentiation (Figure 4A) [45]. To repair dentin–pulp complex damage, DPC proliferation/differentiation and reparative dentin formation are important [45]. Cho et al. found that 3 days after cavity preparation of a murine first molar, DPCs in the cavity prepared tooth were balloon shaped, an indication of DPC odontogenic differentiation while the DPCs in the non-cavity prepared tooth were spindle shaped [45]. The balloon-shaped DPCs deposited reparative dentin after 8 to 10 days [45]. Regarding the expression of autophagy-related genes during odontogenic differentiation, Cho et al. found a less than two-fold change in mRNA expression levels at 3 days post cavity preparation and a greater than two-fold change in mRNA expression levels at 8 days post cavity preparation. Their study suggested that the ATG8 family of genes may be particularly expressed since *GABARAPL2*, *GABARAP*, *MAP1LC3A*, and *MAP1LC3B* were upregulated and *MAP1LC3C* exhibited a significant upregulation in DPCs [45]. Overall, Cho et al. posited that autophagy participates in DPC odontogenic differentiation and proliferation [45].

Autophagy has a role in regenerating dentin–pulp tissue in early caries [46]. In the presence of lipopolysaccharide (LPS), autophagy was enhanced in odontoblasts and autophagy proteins exhibited an increase in expression (Figure 4B) [46]. These effects may be due to the inhibition of the NF-kB pathway since the inhibition of NF-kB facilitated odontoblast differentiation and collagen formation and secretion in odontoblasts [46].

In dental pulp stem cells (DPSCs) treated with interleukin-37 (IL-37) for 3 and 7 days, the Li group found that DPSCs increased expression for the osteoblastic/odontoblastic markers DSPP and RUNX2 [47]. They also found that IL-37-treated DPSCs employed autophagy since autophagy markers were present in the cells as evidenced through Western blotting; specifically, Beclin1 and the ratio of LC3 II/LC3 I were upregulated while P62 was downregulated [47]. When IL-37-treated DPSCs were treated with rapamycin, a molecule that stimulates autophagy, these cells exhibited a greater upregulation of Becilin1 and the ratio of LC3 II/LC3 I and a downregulation of p62 when compared to the IL-37-treated DPSCs [47]. These results indicated that autophagy enhances the osteogenic/odontogenic differentiation of IL-37-treated DPSCs [47].

Within long-lived cells (e.g., odontoblasts, neurons, and cardiomyocytes), autophagy is an important process that facilitates cell survival via replenishment or destruction organelles [48]. When autophagic activity is reduced, lipofuscin (a molecule that indicates odontoblast age, decreases lysosomal activity, and decreases autophagic activity) accumulates in long-lived cells (Figure 4A,C) [48]. Couve and Schmachtenberg observed large autophagic vacuoles in odontoblasts within the dentin–pulp complex and noted that the age of odontoblasts determined the autophagic vacuole content, size, and location [48]. They argued that autophagy serves as a survival mechanism in odontoblasts. Couve and Schmachtenberg suggested that as autophagic activity decreases, lipofuscin granules accumulate in odontoblasts (Figure 4A). The researchers concluded that decreased autophagy and an accumulation of lipofuscin granules are criteria that determine whether human odontoblasts are “old”, outside of chronological age [48]. Additionally, as odontoblasts age, their ability to secrete dentin decreases in response to injury [48]. Couve and Schmachtenberg concluded that autophagy facilitates the dentin-secretion function as odontoblasts age [48].

CPNE7, a transcription factor secreted by pre-ameloblasts that regulates *Dspp* expression in pre-odontoblasts, induces autophagy in odontoblasts [49]. Mature odontoblasts that experienced CPNE7-induced autophagy exhibited a large decrease in lipofuscin when compared to mature odontoblasts that underwent rapamycin-induced autophagy [49]. The removal of lipofuscin, via CPNE7-induced autophagy, reverted the mature odontoblasts back to the physiological state of active secretory odontoblasts (Figure 4A) [49]. In pre-odontoblasts and mature odontoblasts, Park et al. found that CPNE7-induced autophagy increased the amount of odontoblast differentiation and mineralization marker proteins and formed odontoblastic processes [49]. These findings suggest that CPNE7-induced autophagy reactivates mature odontoblasts and stimulates dentin formation, which could aid in dentinal defect therapies [49]. CPNE7-induced autophagy has the potential to reactivate other long-lived cells (e.g., neurons and cardiomyocytes) and treat neurological diseases [49].

**Figure 4 ijms-26-06278-f004:**
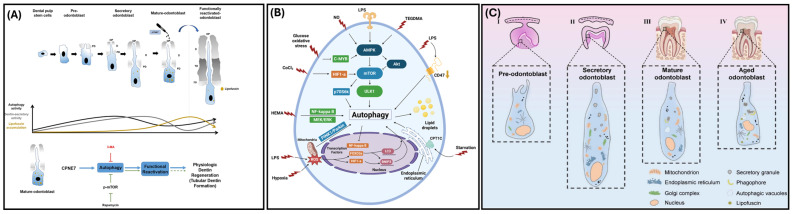
Autophagy in odontoblasts. (**A**) Depiction of odontoblast maturation in relation to autophagic activity and lipofuscin accumulation (reprinted with permissions from Park et al., 2021 [49]). (**B**) Conditions that induce autophagy and the regulatory pathways of autophagy in the dentin–pulp complex (adapted and reprinted with permissions from Yang et al., 2021 [7]). (**C**) Phases of human odontoblast maturation that demonstrate an increase in lipofuscin molecules and a decrease in mitochondria as odontoblasts mature (reprinted with permissions from Meng et al., 2024 [50]).

## 5. Secretory Autophagy

Autophagy, as a degradation/intracellular recycling system, has been shown to play a crucial role in matrix mineralization. Since biomineralization is a carefully regulated process that depends on materials secreted into the extracellular milieu, the role of secretory autophagy must also be explored. One of the most characterized functions of secretory autophagy is its role in immune response cytokine signaling, specifically IL1β [5,11,51]. Lysosomal damage detected by LGALS8 triggers inflammasome activation and subsequent selection of IL1β by TRIM16 [51,52]. Increases in UPS via secretory autophagy can result from stress signals within the cells, particularly in the case of lysosomal dysfunction as seen with IL1β [51,53]. While the exact mechanisms responsible for controlling the decision between degradative or secretory autophagy, its close association with ER machinery and response to stress suggest a role for secretory autophagy in mineralization.

### 5.1. Amphisomes

Amphisomes are intracellular vesicles that occur from the intersection of the autophagic and endocytic pathways [5]. These vesicles arise from fusion of autophagosomes and multivesicular bodies (MVBs)/late endosomes that can fuse with lysosomes for degradation or be secreted (Figure 5) [5]. Several proteins that are critical to exosome biogenesis are also implicated in the formation of amphisomes including SNAREs, Rab GTPases, and endosomal sorting complexes required for transport (ESCRT) [5,54]. SNARE proteins are required for the fusion of the vesicles to the plasma membrane; Syntaxin 17 (STX17) is an autophagosomal SNARE protein that specifically localizes to the outer membrane of autophagosomes. This has been shown to be a necessary step for fusion to late endosomes [55]. The STX17/SNAP29 complex on the autophagosome binds to VAMP8 on the endosomal membrane [55]. Assembly of the STX17/SNAP29/VAMP8 complex is facilitated by Rab7 GTPase interactions with lysosome-related organelles; however, a reduction in Rab7 expression at the transcriptional level increases amphisome formation by weakening the lysosomal degradation pathway [5].

The process of packaging calcium and phosphate in vesicles and secreting them to the ECM is required for matrix mineralization. It has been shown that there is an intricate relationship between amphisome and extracellular vesicle/exosome secretion, particularly where induction of autophagy leads to a decrease in exosome release [5]. However, both vesicles are dependent on the same protein families and machinery (ESCRT and SNARE as seen in Figure 5) indicating that there is a complex regulatory network associated with their compensatory functions [5,54]. With the known role that vesicles play in providing the materials for mineralization, the amphisome–EV relationship provides interesting potential for further understanding the mechanisms of mineralization and how it is influenced by autophagic processes [26].

### 5.2. Secretory Cargo

Intracellular vesicles, including MVBs, autophagosomes, and amphisomes, have highly variable cargo with selection mechanisms that are not entirely understood. Osteogenic differentiation and matrix mineralization both result in altered cargo within various types of vesicles and here we aim to demonstrate that secretory autophagy has the potential to involve components for a mineralizing ECM based on previously described cargo. We briefly highlighted the role that secretory autophagy plays in immune cytokine signaling via IL1β secretion, but this also includes IL6, IL8, and IL18 [5]. Though interleukin cytokine secretion may not directly affect osteogenesis and mineralization, studies have shown that the increased autophagic processes that results in an inflammatory response can simultaneously alter cargo of exosomes [31,56]. A study by Kimura et al. was able to show that ferritin, an iron storage protein, is secreted in the same UPS dependent manner as IL1β [52]. Ferritin is associated with activation of autophagy, but its depletion also results in decreased autophagic functions suggesting that the selection of secretory autophagy cargo may be a part of a self-regulating feedback loop [11,57]. This is further supported by stress-induced secretion of both pro-inflammatory (IL1β/IL8) and anti-inflammatory (ferritin/TGFβ1) signaling molecules via autophagic processes [11].

An in-depth study of an autophagy deficient cell line (Atg5 KO) was able to characterize an autophagy-related secretome that contained a wide array of proteins including, TGFβ1, MMP9, and vimentin [5,58,59]. These results highlighted the diverse cargo of secretory autophagy with functions ranging from anti-inflammatory signaling (TGFβ1) to cell migration (Vim) to ECM degradation and tissue remodeling (MMP9) [5,59].

Glucose regulatory protein 78, also known as HSP5A/BiP, is a molecular chaperone with a vital role in ER stress response regulation and extracellular calcium binding [60]. To maintain its localization, GRP78 has a C-terminal KDEL sequence that binds to KDEL receptors in the ER/Golgi complex [61]. However, without an N-terminal leader sequence, when GRP78 is not bound it is dependent on unconventional secretion to perform its extracellular functions [61]. Another protein that lacks a signal peptide is transforming growth factor beta receptor II interacting protein 1 (TRIP1), a positive regulator of TGFβ signaling [62]. TRIP1 has been found to be exported to the ECM through exosomes (CD63+), however, the interconnecting mechanisms of exosome, autophagosome, and amphisome biogenesis make it possible to have CD63+ secreted autophagic vesicles [62,63].

Along with protein export, secretory autophagy may also have a role in transporting mineral or ion cargo to the ECM such as calcium and phosphate for mineralization. A study on bone homeostasis was able to show that autophagic vesicles contained immature needle-like mineral structure and when autophagy was inhibited the osteoblast function decreased [15]. Additional investigation of osteoblast intracellular vesicles with analytic electron microscopy confirmed the presence of amorphous calcium phosphate particles [26]. Previous publications have shown that secretory autophagy can participate in protein, signaling molecule, and mineral secretion to the ECM and will further explore how this autophagic mechanism contributes to matrix mineralization.

### 5.3. Secretory Autophagy and Matrix Mineralization

Studies included thus far have explored the array of molecules that are associated with secretory autophagy and how these molecules are involved in matrix mineralization. Aside from regulation the ER unfolded protein response pathway (UPR), GRP78 also functions in the ECM during mineralization to bind calcium and aid in calcium phosphate nucleation on collagen fibrils [64]. Increased expression of GRP78 has been shown during osteoblast differentiation which would leave more GRP78 free from the ER and thus available for export to the ECM in a vesicle potentially of autophagic origin [64]. TRIP1 is a component of bone ECM where it can bind type I collagen to support mineralization, and its expression fluctuates during osteo- and odontogenic differentiation [62]. The study by Ramachandran et al. confirmed the export of TRIP1 in CD63+ vesicles which now supports the potential for TRIP1 secretion to be autophagy dependent [62,63]. Knockdown of TRIP1 negatively impacts both differentiation and mineralization supporting that the relationship between secretory autophagy and biomineralization is an interconnected network with intra and extracellular effects [62]. A study by Yuan et al. on the role of IL-17A on osteoblast differentiation found that stimulation with IL-17A induced ferritin expression along with proteins associated with autophagosome formation [57]. They observed an increased in osteogenic differentiation with IL-17A stimulation, and based on their results supported by other findings, determined that this increase could be positively regulated by autophagy activation and ferritin [57]. Increased osteoblast differentiation and support of mineralization would arise not just from autophagy signaling an increase in autophagosomes transporting minerals to the ECM [57].

During the formation of mineralized tissues, calcium and phosphate are packaged within vesicles that will be used to synthesize hydroxyapatite in the ECM [26]. It was previously stated that these vesicles have been identified to be of autophagic origin in osteoblasts [15,26]. These studies of autophagy also found a 50% reduction in bone mass and function loss of the mineralization enzyme TNAP upon inhibition but an increase in mineralization with autophagy stimulation [15,65,66]. From these results it is evident that while secretory autophagy contributes to providing the ECM with both machinery and components for mineralization it is tightly linked to the overarching influence of autophagic signaling on cells and subsequent matrix mineralization.

Bone remodeling and maintenance is dependent on functional osteoblast–osteoclast coupling, a relationship already shown to be affected by autophagy. TIMP1 and MMP9 are both ECM remodeling proteins that have been implicated as autophagosome/amphisome cargo [5]. As TIMPs are classically known to inhibit MMPs, we see further evidence of secretory autophagy implicated in regulation of mineralization processes [67]. The selection of these proteins is influenced by stress response as the study by Martinelli et al. demonstrated that prolonged stress led to lysosomal damage and stimulates the secretory pathway of autophagy (with increased MMP9 secretion and BDNF cleavage in the ECM) [5,59]. As we have established throughout this review, autophagic processes are integral to matrix mineralization which is further supported in a multifactorial way by secretory autophagy pathways.

## 6. Clinical Implications

Autophagic processes are vital for cell survival and proper maintenance of homeostasis, thus changes to its regulatory mechanisms or mutations in key molecules can result in a variety of pathologies. Understanding the role that autophagy plays in the development and progression of these diseases is critical to the design of new treatments. Rheumatoid arthritis (RA) is an autoimmune disease characterized by chronic joint inflammation that can lead to irreversible destruction of bone and cartilage [68]. The exosomes produced in the joints of RA patients have been shown to promote disease progression, however, the environment of chronic inflammation and stress increase the autophagy response of these cells which could subsequently influence exosome biogenesis [68]. Further investigation of how autophagy and exosome production in this context are connected could provide a new axis for therapies to limit/slow down disease progression. Studies on changes to odontoblasts in an environment altered by LPS stimulation show an increase in autophagy resulting in both an increase in cell differentiation and a decrease in cell viability [46]. Characterization of the mechanisms associated with the opposing effects of autophagy on these cells may present an interesting avenue for the repair of caries [69].

Autophagy also has implications in pathologies such as vascular calcification. As autophagic processes are shown to play a role in the differentiation of osteoblasts, studies have shown that it can stimulate this differentiation in vascular smooth muscle cells [33]. Changes in calcium signaling due to cardiac diseases can alter autophagic signaling and machinery that contributes to vascular calcification [33]. Further elucidation of how this pathological process occurs would provide valuable insight and potential autophagy-based therapies for this condition.

Aside from correcting deficits with autophagy signaling, its machinery and pathways could also be used for repair or regeneration of defects in mineralized tissues. Autophagosomes present a promising candidate for tissue engineering even though there are currently no clinical trials specifically targeting the regulatory aspect of secretory autophagy [11]. There are many studies that look to treat acute and chronic conditions using engineered extracellular vesicles (via microRNA/non-coding RNA cargo) and with continued investigation of secretory autophagy cargo selection it could be possible to alter endogenous cell processes through autophagy for similar applications [11,46].

## 7. Conclusions

Autophagy affects matrix mineralization, during the formation of bones and teeth. In mineralization, autophagy facilitates osteoblastic and odontoblastic differentiation and synthesis of a mineralized matrix. In bone, autophagy influences osteoblast, osteoclast, and osteocyte activity and thus affects bone remodeling. In teeth, autophagy participates in odontogenic differentiation and facilitates odontoblastic primary and secondary dentin secretion. This review highlights the effects of various components in both autophagy and mineralization and the studies explored show that autophagy is integral to bone and dentin mineralization by supporting intracellular signaling pathways required for cell differentiation and subsequent matrix mineralization.

## 8. Materials and Methods

A scientific literature search was used as the basis for writing this review article. Various combinations of the keywords autophagy, odontoblast, macroautophagy, secretory autophagy, dentin, biomineralization, matrix mineralization, tooth, dental pulp stem cell, osteoblast, calcification, tooth calcification, osteogenesis, and odontogenesis were used in PubMed to elucidate previous data and findings in the field. Many studies directly demonstrated the importance of autophagy in cell populations that form mineralized tissues. Our search allowed us to synthesize important studies and highlight the role of a specific cellular process, across a variety of cell populations, and how it relates to matrix mineralization.

## Figures and Tables

**Figure 1 ijms-26-06278-f001:**
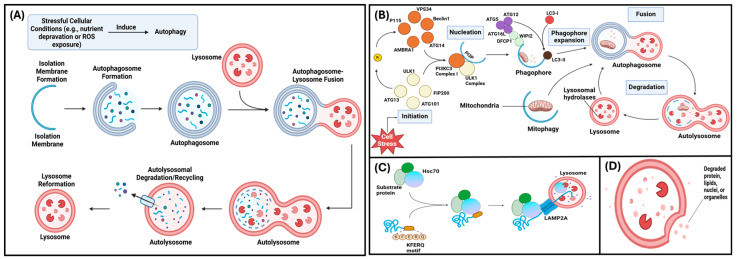
Various modes of autophagy—(**A**) Pathway demonstrating overall mechanism of autophagy. (**B**) Schematic of macroautophagy. (**C**) Schematic of chaperone-mediated autophagy. (**D**) Diagram of microautophagy (adapted and reprinted with permissions from Yang et al., 2021 [7]).

**Figure 5 ijms-26-06278-f005:**
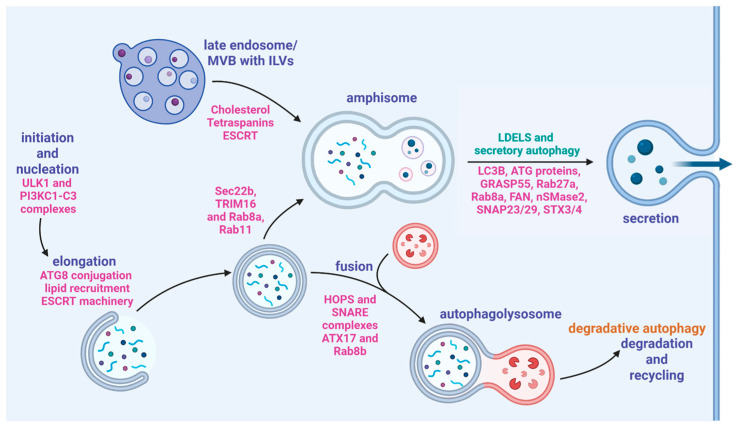
Secretory autophagy pathways demonstrating the steps and types of vesicles that can be formed with the important proteins involved in each step. Autophagosome formation (far left) can be followed by fusion with a lysosome (bottom) or fusion to a late endosome (top) (reprinted with permissions from Weigert et al., 2024 [11]).

## Data Availability

No new data were created or analyzed in this study. Data sharing is not applicable to this article.

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
