# Peer review of "The Role of Autophagy in the Mineralization Process of Bone and Dentin"

_ijms, 2025, doi:10.3390/ijms26136278_

Round 1
Reviewer 1 Report
Comments and Suggestions for Authors
In their review, Moran and colleagues aim to elucidate the significance of autophagy during matrix mineralization. The manuscript could be of interest; however, several drawbacks need to be addressed. The title should be modified to clearly indicate that the review exclusively discusses matrix mineralization in bones and teeth, without reference to other physiological or pathological contexts. Furthermore, the text mainly focuses on the role of autophagy in the differentiation of osteoblasts, osteoclasts, and odontoblasts, as well as on the characteristics of secretory autophagy. The relationship between autophagy and matrix mineralization is not sufficiently emphasized. For instance, it does not explain how the ions involved in matrix mineralization are packaged within cell-derived vesicles, nor whether such vesicles originate from secretory autophagy or they are matrix vesicles or exosomes. Additionally, all the images are reprinted from previously published studies, and no original figures are provided. For these reasons, as well as others detailed below, the review is not suitable for publication in its current form.
Detailed comments
- Line 2: The title does not clearly indicate that the review is exclusively focused on bone and tooth tissues. Moreover, the title is inconsistent with the review’s content, which primarily addresses the role of autophagy in cell differentiation within bones and teeth
- Lines 56, 116-119, 187-190, 258-262, and 303-304: All the images are reprinted from previously published studies. This is unconventional. Tipically, one or two original graphical images should be included
- Figure 2: The content in Figure 2B regarding bone structure is incorrect. The Haversian canal and bone lamellae are incorrectly indicated by arrows. The collagen fiber does not show any banding, which is typical of collagen fibrils; instead, the image resembles a skeletal muscle fiber. In the panel below, “APC deposition in the collagen fibril” should be revised to “APC deposition on collagen fibrils” or “between collagen fibrils”, as the image depicts collagen interfibrillar mineralization
- Lines 95 and 354: Collagens should be referred to using Roman numerals (collagen I, II, III, etc.), not Arabic numerals (collagen 1, 2, 3, etc.)
- Lines 121-143: The effects of ion-mediated autophagy are unclear. This paragraph needs to be improved
- Line 343: The phrase “We’ve shown that…” is inappropriate, as the review summarizes data from other studies rather than presenting original research
- Lines 347-348: The expressions “Thus far we have explored…” and “…aim to demonstrate how…” are also unsuitable for the same reason (see comment above)
Author Response
Thank you for your comments and suggestions. We have highlighted changes throughout the text in red and included responses to each comment listed below.
Comment 1: Line 2: The title does not clearly indicate that the review is exclusively focused on bone and tooth tissues. Moreover, the title is inconsistent with the review’s content, which primarily addresses the role of autophagy in cell differentiation within bones and teeth
Response 1: This review aims to discuss the intersection of autophagy and mineralization in the context of bone and teeth where cell differentiation is an important step in the processes. As we are not focused specifically on bones or teeth the title refers to matrix mineralization to cover these different locations as well as encompass the entire process not just cell differentiation, mineral transport or deposition.
Comment 2: Lines 56, 116-119, 187-190, 258-262, and 303-304: All the images are reprinted from previously published studies. This is unconventional. Typically, one or two original graphical images should be included
Response 2: Fig 1A is an original figure and there are several sections of other figures that have been remade for the purpose of this review. We included citations for each figure as ‘adapated’ to ensure that proper credit is given despite any minor alterations.
Comment 3: Figure 2: The content in Figure 2B regarding bone structure is incorrect. The Haversian canal and bone lamellae are incorrectly indicated by arrows. The collagen fiber does not show any banding, which is typical of collagen fibrils; instead, the image resembles a skeletal muscle fiber. In the panel below, “APC deposition in the collagen fibril” should be revised to “APC deposition on collagen fibrils” or “between collagen fibrils”, as the image depicts collagen interfibrillar mineralization
Response 3: The arrows have been adjusted correctly label the bone structure and the collagen fiber does show banding. In the panel below the syntax for APC deposition has also been adjusted.
Comment 4: Lines 95 and 354: Collagens should be referred to using Roman numerals (collagen I, II, III, etc.), not Arabic numerals (collagen 1, 2, 3, etc.)
Response 4: Thank you, this has been adjusted in the text.
Comment 5: Lines 121-143: The effects of ion-mediated autophagy are unclear. This paragraph needs to be improved
Response 5: The material presented in this section aimed to give a brief overview of how ions involved in mineralization can be influenced by autophagy; we have added some additional material on the role of Mg2+ and clarified that with the presented studies we are summarizing the importance of these ions.
Comment 6: Line 343: The phrase “We’ve shown that…” is inappropriate, as the review summarizes data from other studies rather than presenting original research
Response 6: This phrase has been adjusted to reflect that other work is being summarized.
Comment 7: Lines 347-348: The expressions “Thus far we have explored…” and “…aim to demonstrate how…” are also unsuitable for the same reason (see comment above)
Response 7: These phrases have been adjusted to reflect that other work is being summarized.
Reviewer 2 Report
Comments and Suggestions for Authors
The Introduction quickly orients non-specialists by defining autophagy, outlining its three major forms, and linking them to mineralization before delving into specifics. Unlike many bone-centric reviews, this article devotes substantial space to dentin and cementum, offering readers a fuller picture of autophagy-guided hard-tissue formation. These strengths mean that, once minor structural and methodological revisions are addressed, the manuscript should provide a thorough and engaging resource for both newcomers and specialists interested in the autophagy–mineralization nexus.
Major concerns
1. The Abstract currently contains sentences describing the authors’ literature-selection strategy (e.g., database name, search terms, inclusion criteria). While transparency is important, detailed methodological descriptions belong in the main text—typically the Introduction (or a dedicated Methods subsection for narrative reviews)—not in the Abstract.
2. A disproportionately large fraction of cited papers come from the senior author’s own group and from a few recurring laboratories. That can skew conclusions; widen the citation base. The review barely mentions adverse or paradoxical roles of autophagy in pathological calcification (e.g., vascular media calcification, pulp calcifications) or the effects of autophagy inhibitors—omitting a critical body of counter-evidence. The newest citations stop in early 2024, yet several high-impact 2024-2025 papers on mitophagy-driven osteoblast dysfunction and secretory-autophagy assays are absent. Stating the search end-date would clarify this.
Minor concerns
1. "Mineralization is the cellular process …” is stated twice (lines 83–90 and again 88–90).
2. The paper repeatedly asserts that autophagy is “critical” or “vital” for mineralization (e.g., Abstract lines 20-21) without acknowledging conflicting data or specifying context.
3. The text introduces ~35 abbreviations; several (e.g., UPS, TRIP1) appear only once. Prune or spell out on first use and consider an abbreviation table.
4. Lines 121-136 (Mg²⁺ section). The antagonistic role of Mg²⁺ in ECM mineralization is oversimplified; low-Mg diets can increase bone fragility even when calcium deposition is normal. Add nuance and cite contrasting data.
5. Lines 168-184 (osteocyte Atg7 KO). The review cites Onal et al. for male-female differences but does not discuss the potential hormonal confounders (oestrogen signalling influences autophagy).
Examples include “An subsequent effect” (line 97), “engendered” used repeatedly, and inconsistent pluralisation of osteoblast/odontoblasts. A careful language edit is needed.
Author Response
Thank you for your comments and suggestions. We have highlighted changes throughout the text in red and included responses to each comment listed below.
Comment 1: The Abstract currently contains sentences describing the authors’ literature-selection strategy (e.g., database name, search terms, inclusion criteria). While transparency is important, detailed methodological descriptions belong in the main text—typically the Introduction (or a dedicated Methods subsection for narrative reviews)—not in the Abstract.
Response 1: Thank you, these details were already included in the text and have been removed from the abstract.
Comment 2: A disproportionately large fraction of cited papers come from the senior author’s own group and from a few recurring laboratories. That can skew conclusions; widen the citation base. The review barely mentions adverse or paradoxical roles of autophagy in pathological calcification (e.g., vascular media calcification, pulp calcifications) or the effects of autophagy inhibitors—omitting a critical body of counter-evidence. The newest citations stop in early 2024, yet several high-impact 2024-2025 papers on mitophagy-driven osteoblast dysfunction and secretory-autophagy assays are absent. Stating the search end-date would clarify this.
Response 2: The bulk of the search was completed at the end of 2024, with some additional being added through editing processes, however we expanded the search during the revision process to include publications from 2025.
Minor concerns
Comment 3: "Mineralization is the cellular process …” is stated twice (lines 83–90 and again 88–90).
Response 3: The first occurrence has been removed as to avoid repetition.
Comment 4: The paper repeatedly asserts that autophagy is “critical” or “vital” for mineralization (e.g., Abstract lines 20-21) without acknowledging conflicting data or specifying context.
Response 4: The phrasing in the abstract has been changed and occurrences throughout the paper were altered directly or placed with additional context.
Comment 5: The text introduces ~35 abbreviations; several (e.g., UPS, TRIP1) appear only once. Prune or spell out on first use and consider an abbreviation table.
Response 5: We have generated an abbreviation table that was submitted as supplemental material and ensured that they are spelled out on first use.
Comment 6: Lines 121-136 (Mg²⁺ section). The antagonistic role of Mg²⁺ in ECM mineralization is oversimplified; low-Mg diets can increase bone fragility even when calcium deposition is normal. Add nuance and cite contrasting data.
Response 6: Thank you for this observation, we have supplemented this section to further clarify this concept.
Comment 7: Lines 168-184 (osteocyte Atg7 KO). The review cites Onal et al. for male-female differences but does not discuss the potential hormonal confounders (oestrogen signalling influences autophagy).
Response 7: We supplemented the point made here with additional studies that address the link between male/female differences as they relate to autophagy and bone maintenance.
Comments on the Quality of English Language
Comment 8: Examples include “An subsequent effect” (line 97), “engendered” used repeatedly, and inconsistent pluralisation of osteoblast/odontoblasts. A careful language edit is needed.
Response 8: The above examples have been corrected and the text has been thoroughly checked for additional language edits.
Round 2
Reviewer 1 Report
Comments and Suggestions for Authors
The review by Moran and colleagues has been improved through revision. However, two aspects still require attention:
First, the title does not accurately reflect the scope and content of the review, as it implies to a broader context than is actually addressed.
Second, Figure 2B continues to propagate a misleading concept, as it depicts the collagen fiber as a banded structure, whereas it is the collagen fibril that exhibits this characteristic.
Author Response
Comment 1: First, the title does not accurately reflect the scope and content of the review, as it implies to a broader context than is actually addressed.
Response 1: We have adjusted the title to be more specific and better reflect the content of the review
Comment 2: Second, Figure 2B continues to propagate a misleading concept, as it depicts the collagen fiber as a banded structure, whereas it is the collagen fibril that exhibits this characteristic.
Response 2: Thank you for clarifying this point, we understand that the collagen fiber in its previous state strongly resembled a muscle fiber. Adjustments have been made to the figure to more accurately depict the structural hierarchy.
Reviewer 2 Report
Comments and Suggestions for Authors
Authors addressed all issues raised by reviewer. I don’t have further comments on this article.
Author Response
No additional comments made from Reviewer 2